# Status Update: Is smoke on your mind? Using social media to assess smoke exposure

Bonne Ford[1], Moira Burke[2], William Lassman[1], Gabriele Pfister[3], and Jeffrey R. Pierce[1]

[1]Department of Atmospheric Science, Colorado State University, 1371 Campus Delivery, Fort

5    Collins, CO 80523

[2]Facebook, Menlo Park, CA 94025

[3]National Center for Atmospheric Research, 3450 Mitchell Lane, Boulder, CO 80301

*Correspondence to*:  Bonne Ford ([bonne@atmos.colostate.edu](mailto:bonne@atmos.colostate.edu))

**Abstract.**

Exposure to wildland-fire smoke is associated with negative effects on human health. However, these effects are poorly quantified. Accurately attributing health endpoints to wildland-fire smoke requires determining the locations, concentrations, and durations of smoke events. Most current methods for assessing these smoke events (ground-based measurements, satellite observations, and chemical-transport modeling) are limited temporally, spatially, and/or by their level of accuracy. In this work, we explore using daily social-media posts regarding smoke, haze, and air quality from Facebook to assess population-level exposure for the summer of 2015 in the western US. We compare this de-identified, aggregated Facebook data to several other datasets that are commonly used for estimating exposure, such as satellite observations (MODIS aerosol optical depth and Hazard Mapping System smoke plumes), daily (24-hour) average surface particulate-matter measurements, and model (WRF-Chem) simulated surface concentrations. After adding population-weighted spatial smoothing to the Facebook data, this dataset is well-correlated ($R^2$ generally above 0.5) with these other methods in smoke-impacted regions. The Facebook dataset is better correlated with surface measurements of $PM_{2.5}$ at a majority of monitoring sites (163 of 293 sites) than the satellite observations and our model simulation are. We also present an example case for Washington state in 2015, where we combine this Facebook dataset with MODIS observations and WRF-Chem simulated $PM_{2.5}$ in a regression model. We show that the addition of the Facebook data improves the regression model's ability to predict surface concentrations. This high correlation of the Facebook data with surface monitors and our Washington state example suggests that this social-media-based proxy can be used to estimate smoke exposure in locations without direct ground-based particulate-matter measurements.

## 1 Introduction

Exposure to poor air quality is associated with negative impacts on human health (Dockery et al., 1993; Pope, 2007). As such, the Environmental Protection Agency (EPA) has set air-quality standards to limit concentration levels of pollutants in the United States, which has led to reductions in anthropogenic emissions. However, particulate matter (PM) also has natural and transboundary sources, which are more difficult to control. A large natural source of PM in the western US is from landscape fires, which are comprised of wildfires and prescribed burning on natural lands and agricultural fires. Landscape fire smoke (LFS) drives much of the

interannual variability in total $PM_{2.5}$ (PM with an aerodynamic diameter < 2.5 µm, Jaffe et al., 2008). The 2011 National Emission Inventory (NEI2011, epa.gov) attributes ~20 % of the primary $PM_{2.5}$ emissions in the US to wildfires, 15 % to prescribed fires, and 1.5 % to agricultural fires (epa.gov). Lelieveld et al. (2015) used concentration-response functions derived

from previous studies of total ambient PM (and smoking and household air pollution) to estimate that ~2500 premature mortalities are attributable to exposure to biomass-burning (a broad category that includes wildland, prescribed, and agricultural fires) $PM_{2.5}$ per year in the US. However, the assumed toxicity and dose associated with LFS were assumed to be the same as all other PM sources. Thus, it is important to determine the health responses specific toLFS.

Accurately attributing health outcomes to LFS requires a determination of the exposed population. Studies of health impacts often rely on (I) fixed-site monitors (e.g. Pope et al., 2009), (II) satellite products (e.g. Henderson et al., 2011; Rappold et al., 2011), , or (III)atmospheric model simulations (Alman et al., 2016; Fann et al., 2012; Johnston et al., 2012; Rappold et al., 2012),. Each of these methods has limitations as an exposure metric. For

example, fixed site monitors are sparse in much of the western US, and satellite products do not on their own provide surface-level concentrations. Atmospheric model simulations may be biased by their emission inventories (Davis et al., 2015; Zhang et al., 2014), spatial resolution (Misenis and Zhang, 2010; Punger and West, 2013; Thompson et al., 2014; Thompson and Selin, 2012), or input meteorological fields (Cuchiara et al., 2014; Srinivas et al., 2015; Žabkar et al.,

2013). Thus, there is a growing effort to include multiple datasets (e.g. Henderson et al., 2011; Yao et al., 2013) and create blended products that can exploit the strengths of each dataset (Brauer et al., 2015; van Donkelaar et al., 2015; Lassman et al., 2017; Reid et al., 2015; Yao and Henderson, 2013). However, these methods still only provide estimates of ambient concentration levels and not of actual exposure. Additionally, attributing health effects specifically to LFS

exposure can be difficult as it requires separating the contribution of smoke from total $PM_{2.5}$ (Liu et al., 2015).

   In this work, we propose the use of de-identified, aggregated Facebook data to determine population-level exposure for the summer of 2015, which was a particularly smoky year in the US (See Supplementary Figure 1 for number of fire and smoke days). While there can be many

different sources of poor air quality, the highest $PM_{2.5}$ concentrations measured during the study period were in regions and during time periods associated with wildfire smoke. We show that,

region-wide, this dataset is better correlated with surface measurements of $PM_{2.5}$ than other traditional means of estimating exposure, suggesting that it has the potential to be used to estimate smoke exposure in locations without direct ground-based particulate-matter measurements. We also present a test case for Washington state, where we demonstrate that a

regression model that includes our Facebook dataset is better able to predict surface $PM_{2.5}$ than a regression model that only has model-simulated $PM_{2.5}$ and satellite aerosol optical depth (AOD). We also compare our results to another measurement of internet behavior, Google Trends, as a proxy for air-quality exposure.

The use of social media in risk and exposure assessment is a growing field. In the past

decade, data mining of social media has provided a wealth of information to news outlets, marketing firms, and the social sciences (Burke and Kraut, 2016; Golder and Macy, 2011; Kosinski et al., 2013; Masedu et al., 2014; Youyou et al., 2015). Only recently have social media and internet behavior been used for research in both the natural sciences and public health. Social media and internet behavior have been proposed to track epidemics and earthquakes (e.g.

Broniatowski et al., 2013; Crooks et al., 2013; Ginsberg et al., 2009),  fires (Abel et al., 2012; Bedo et al., 2015; De Longueville et al., 2009; Kent and Capello Jr, 2013), and poor air quality (Jiang et al., 2015; Mei et al., 2014; Tao et al., 2016), and to predict hospitalizations (Ram et al., 2015). A paper by Sachdeva et al. (2016) also proposed the use of Twitter content and geographic information to estimate LFS concentrations. In this paper, we show how daily

Facebook posting trends "track" significant changes in air quality, such as is associated with dense smoke plumes from large wildfires. Furthermore, we show that Facebook posting trends could also improve estimates of $PM_{2.5}$ exposure by serving as an extra constraint on more traditional methods for estimating exposure.

**2    Methods and Datasets**

**2.1 Internet Behavior Datasets**

**2.1.1 Aggregate Percent of Facebook Posters**

Our dataset is the percentage of distinct Facebook posters in each US city that used any of the following words: "smoke", "smoky", "smokey", "haze", "hazey", or "air quality" in a

post, while attempting to filter out reference to cigarette smoking and other phrases not related to air quality (see Supplement). The search generates de-identified and aggregated counts of posters

each day, divided by the number of people who used Facebook in that city. This method counts each person at most once per day, avoiding bias from a single person posting multiple times about air quality that day. Re-shares of news articles and friends' posts were also not included. No individual's text was viewed by researchers. Our goal was to focus on wildfire smoke

because wildfire smoke often leads to extreme air quality degradation over broad regions of the US in the summertime. However, because this list includes "air quality" and "haze" (and results were all aggregated), this search criterion can also highlight trends in Facebook posters discussing air quality degradation due to other emissions, such as fossil-fuel combustion, and may better encompass more of the ways that people discuss their experiences of changes in the

air form smoke or other particulate matter. Geographic location at the city level is determined by IP address. Data were provided for 5 June through 27 October 2015.

We analyzed this dataset of the de-identified, aggregated percent of Facebook posters that matched our search criteria at the city, town, or other municipality level (See Supplementary Figure 2a for location centroids, referred to as "raw" throughout text). We translate the percent

of Facebook posters data on to a standard latitude/longitude grid using an area-smoothing procedure with data weighted by the population of the municipality (See Supplementary Figure 2 for example). The spatial interpolation allows us to estimate the magnitude of the response between the specific locations (centroids) and to compare to other gridded datasets. Additionally, we chose to weight the results by population because some of these locations are in areas with

small populations (and potentially few posters on Facebook), which can skew results. We generated a fixed 0.25° grid using an inverse distance weighting to a power of six with a scale distance (or search neighborhood, $d_s$) of 20 km. The scale distance and power were set to sharply reduce the influence of more-distant observations and were chosen based on the grid resolution in order to maintain the regional variability from the Facebook posters. Our resulting gridded

data are determined using the following formula:

$$f_i = \frac{\Sigma\left\{f_c \times \frac{P_c}{\left[1+\left(\frac{d_{i,c}}{d_s}\right)^6\right]}\right\}}{\Sigma\left\{\frac{P_c}{\left[1+\left(\frac{d_{i,c}}{d_s}\right)^6\right]}\right\}} \tag{1}$$

Where the percent of Facebook posters ($f_i$) at a grid location ($i$) is the sum of all of the products of the population ($P_c$) and the original percent of Facebook posters ($f_c$) at each "Facebook municipality" ($c$), weighted by the inverse of the distance ($d$) between location ($i$) and the Facebook municipality ($c$).

**2.1.2 Google Trends**

We also analyzed Google Trends data (google.com/trends/) as a proxy for exposure and to evaluate the keywords used in our search criteria. Our reason for including this analysis is twofold: (1) to compare the results of our percent of Facebook posters comparison to results using another internet behavior dataset and (2) to determine which keywords are most strongly correlated with $PM_{2.5}$ (as our Facebook posters dataset is an aggregated result for all search terms). We searched for "air quality", "wildfire", "smoke", "pollution", "haze", "smog", and "ozone" for 1 May – 31 October 2015 for every designated media area (DMA) in the western US. Google Trends results are determined from a random sample of searches with location determined by IP address and duplicates (when the same person searches for the same term multiple times) removed. Results for each search are aggregated and de-identified, but limited to popular terms, with low values appearing as zero (highest values are 100). Therefore, the popularity of a search term impacts the spatial resolution available of the aggregated results (country, DMA, or city). Because of the coarse resolution of the aggregated Google Trends data (DMA-level), we chose to compare only to surface measurements and not the other gridded datasets. In order to determine the temporal correlation between the Google Trends and surface measurements, we identified the DMA in which each measurement site is located.

**2.2 Surface Measurements**

We determined the temporal correlation of these datasets to several other datasets that are commonly used for estimating exposure to LFS on a daily timescale. We use 24-hour average concentrations of total $PM_{2.5}$ mass from EPA Air Quality System (AQS, data from epa.gov/aqs), which includes monitor data from different agencies, and Interagency Monitoring of Protected Visual Environments (IMPROVE, data from views.cira.colostate.edu/fed/) sites. At IMPROVE network sites, surface measurements of atmospheric composition are taken over a 24-hour period every third day (Malm et al., 1994). Depending on the measurement method at the site, 24-hr average concentrations are provided daily, every third day, or every sixth day at EPA-AQS sites. To maximize our data availability, we are using measurements from Federal Reference Method

and Federal Equivalent Method (FRM/FEM, 88101) sites and from non-FRM/FEM (88502) sites (both are also used by the EPA for AQI summaries).

We determined the temporal correlations between the daily surface measurement and the internet behavior datasets at every site. However, in the Results and Discussion section, we only show example time series for four of these locations. These four locations are shown because they were all impacted by wildfire smoke during the study period, but the response in the percent of Facebook posters varied among the sites likely due to differences in surface concentrations, distance to fire, population, and cloud cover (discussed in Results and Discussion).

**2.3 Satellite Products**

**2.3.1 Hazard Mapping System (HMS) Smoke Product**

We also use the Hazard Mapping System (HMS) fire and smoke analysis product, which is produced routinely by the National Oceanic and Atmospheric Administration's (NOAA) National Environmental Satellite and Data Information Service (NESDIS) for the purpose of identifying fires and smoke emissions (satepsanone.nesdis.noaa.gov). The HMS smoke product uses observations from both geostationary and polar-orbiting satellites. Polygons determined from satellite visible image analysis are currently categorized as light, moderate and heavy smoke and have assigned numerical values to estimate surface smoke concentrations (5, 16, 27 $\mu$g m$^{-3}$). This product is only available for daylight hours and each polygon is considered valid for a specific time period. We created a gridded surface from all the polygons valid for each day with the surface-concentration values suggested at the same 0.25° grid resolution as our gridded percent of Facebook posters in order to calculate the temporal correlation between the two datasets for each grid. In grids where there is more than one polygon valid for a day, we take the maximum value in each grid location during that day. Data files were available for every day during our analysis period except 20 August 2015, although sub-daily smoke plume analysis periods could also be missing. For determining the correlation with surface measurements, we matched the site location to the corresponding grid box.

**2.3.2 MODerate resolution Imaging Spectroradiometer (MODIS) AOD**

For AOD from satellites, we use the Collection 6, MODerate resolution Imaging Spectroradiometer (MODIS) Level 2 10-km aerosol optical depth (AOD) products from the Terra and Aqua platforms. Terra has a morning overpass (~10:30 AM LT) and Aqua has an afternoon overpass (~1:30 PM LT). With a swath width of 2,330 km, the instruments provide

almost daily coverage of the globe in cloud-free conditions. The MODIS algorithm can have difficulty distinguishing thick smoke from cloud (van Donkelaar et al., 2011), causing some instances of heavy smoke to be erroneously filtered out (although Collection 6 has made improvements to the algorithm to minimize this misclassification, see Levy et al., 2013). We

average the MODIS AOD observations from both instruments on the same 0.25° grid and use all quality levels for better coverage. We additionally use the MODIS cloud fraction (CF) products ("Cloud_Fraction_Land" and "Cloud_Fraction_Ocean,") in order to determine the presence of clouds and to determine if cloudiness impacts Facebook postings on smoke. We calculate the temporal calculations between MODIS AOD and the Facebook posters dataset and the surface

observations for the full dataset and excluding cloudy days.

**2.4 Weather Research and Forecasting model with Chemistry (WRF-Chem) PM$_{2.5}$**

Several models and model frameworks are also routinely used to estimate smoke exposure. Here, we use a chemical transport model, the Weather Research and Forecasting model with Chemistry (WRF-Chem). The simulation was completed for 5 June – 1 October

2015. We use Global Forecast System (GFS) meteorology, biogenic emissions from the Model of Emissions of Gases and Aerosols from Nature (MEGAN, Guenther et al., 2006), National Emission Inventory 2011 (NEI) anthropogenic emissions, FINN biomass-burning emissions (Wiedinmyer et al., 2011), MOZCART aerosol species and chemistry, and (MOZART) chemical boundaries (Emmons et al., 2010). Horizontal resolution is 15 km and there are 27 vertical

levels. Concentrations are output for each model hour, which we then average to provide daily 24-hour average surface concentrations in order to compare to the percent of Facebook posters dataset and surface measurements.

**2.5 Regression Model**

We also present a test case to evaluate the feasibility and usefulness of including this

percent of Facebook posters dataset in a statistical model. We compare two geographically weighted regression (GWR) models that use MODIS AOD and WRF-Chem PM$_{2.5}$ with and without the Facebook posters dataset. GWR has previously been used in a several different studies to predict surface air quality (Hu et al., 2013; Lassman et al., 2017; Song et al., 2014; You et al., 2016). For our test case, we focus on Washington state because of the extensive

network of surface PM$_{2.5}$ measurements available for validating results. In our regression model, we determine the dependent variable (surface PM$_{2.5}$ at each measurement site) from a linear

combination of these different predictor variables (MODIS AOD, WRF-Chem PM$_{2.5}$, and gridded percent of Facebook posters). A separate set of regression coefficients is determined at each surface monitor location, which are then interpolated across the domain. We use the leave one out cross validation (LOOCV) method to test our models, in which the regression

coefficients determined at a single monitor are removed from the interpolation scheme, and then the resulting PM$_{2.5}$ predicted by the regression model is compared to the observed PM$_{2.5}$ concentrations. We calculate the temporal correlation, slope, and mean absolute error (MAE) for the two regression models.

**3   Results and Discussion**

**3.1 Comparison of Percent of Facebook Posters to Conventional Metrics**

An example of the data used in this study is given in Figure 1 for 29 June 2015, which shows a dense smoke plume from wildfires in Canada causing degraded air quality over the Midwestern US and smoke from local fires in the Northwest over Washington, Oregon, and

Idaho. The impact of this smoke plume is evident in the HMS smoke product, the anomalously high surface PM$_{2.5}$ concentrations, the elevated MODIS AOD values, and in the WRF-Chem PM$_{2.5}$. The spatial pattern in the percent of Facebook posters is somewhat consistent with regions of degraded surface air quality, suggesting some people were aware of the degraded air quality. The extent of the "Facebook plume" does not extend as far east or as far south as the smoke

plume observed by the satellite products (MODIS AOD and HMS smoke product), and hotspots in the percent of Facebook posters are centered around the eastern Montana/Canada border. To note, the surface measurements also do not show a strong increase in surface concentrations as far south (Missouri and Arkansas), suggesting that the plume observed by the satellites might have been lofted above the surface. Additionally, while the HMS smoke product suggests only

light smoke over northeastern Montana and MODIS AOD  is only moderately higher than the surrounding region, surface PM$_{2.5}$ concentrations are elevated, which agrees with the spatial pattern in Facebook posters. In cases where the plume is lofted or smoke is concentrated at the surface, this new dataset might be more representative of surface air-quality changes than these satellite products.

In Figure 2, we also show example time series of percent of Facebook posters and other datasets (surface PM$_{2.5}$ measurements, MODIS AOD, MODIS CF, HMS smoke product) used in

this study for four different locations in the western US: Fort Collins, CO; Pinehurst, ID; Bellingham, WA; and Great Falls, MT. All four of these locations were impacted by wildfire smoke during the study period, but the response in the percent of Facebook posters varied among the sites likely due to differences in surface concentrations, distance to fire, population, and

cloud cover. From these time series, we see the main two fire event periods that impacted large areas of the US during the summer of 2015: (1) the Canadian wildfires in late June through early July and (2) the wildfires in the northwestern US (mainly Washington and Idaho) in August. The magnitude of impact on these different metrics for estimating air quality varies by location and event. For Pinehurst, ID, where the population was ~1600 in 2015, population-weighting the

Facebook posters time series improves the correlation with the 24-hour average surface measurements ($R^2 = 0.55$ for gridded and $R^2 = 0.00$ for raw). In more populated regions, such as Fort Collins, CO (pop. ~161,000), Bellingham, WA (pop. ~85,000), and Great Falls, MT (pop. ~60,000); population-weighting the Facebook posters has little impact on the time series and resulting correlation with the surface measurements (as shown in Supplement Figure 3). Further

discussion of these time series is presented throughout this result section.

        In order to assess how well changes in the fraction of people posting about smoke and air quality in Facebook posts represent actual changes in surface air quality, we compare time series of the percentage of Facebook posts matching our criteria to time series of $PM_{2.5}$ measured at all of the different surface sites across the summer of 2015, such as shown in the example time

series of Figure 2. The coefficients of determination for all surface $PM_{2.5}$-measurement sites with the gridded, population-weighted Facebook posts are shown in Figure 3a, which suggests that the best agreement between the two datasets is in regions that experienced heavy smoke and/or anomalously high $PM_{2.5}$ concentrations during the summer, which is to be expected based on our search criteria. For example, the Mt. Hood IMPROVE site in Oregon (Figure 3) had 39

measurement days (June 5-September 30) and had 14 days when the HMS smoke product suggested smoke over the location. This site provides the best $R^2$ between the percent of Facebook posters and measured surface $PM_{2.5}$ with a value of 0.97.

        We also compare agreement of the percent of Facebook posters against simulated concentrations from a chemical transport model simulation (WRF-Chem, Figure 3b), which

again shows the highest correlation in the Northwest US, which was impacted by wildfire smoke for many days in the summer of 2015. We would expect this as our Facebook posters search

criteria is aimed at smoke and poor air quality and would likely only show changes in postings in regions where air quality was noticeably degraded.

Agreement between MODIS AOD and Facebook posting trends are shown in Figure 3c, which also shows the best agreement in the northwestern US. Because thick smoke can

occasionally be classified as cloud in the MODIS algorithm (van Donkelaar et al., 2011), we filter out MODIS AOD observations where the cloud fraction was > 75 %. The impact of this filtering is shown in the time series of Figure 2. The criterion reduced our number of useable observations but improved correlations at most sites (Supplementary Figure 4). Comparisons between Facebook posters and MODIS AOD are similar spatially to WRF-Chem $PM_{2.5}$ and

surface measurements, but coefficients for MODIS AOD and Facebook posts are generally worse. However, this satellite product is derived for the full atmospheric column and is not necessarily directly relatable to surface concentrations. Smoke plumes (and transported pollution from other sources) can be lofted above the surface and may not impact surface-level exposure where astute Facebook posters would take notice.

Finally, we also show $R^2$ values between the HMS smoke product estimated values and the Facebook posters in Figure 3d. Again, we see similar trends, where the best agreement occurs in regions which experienced numerous smoke days. The correlation values are not as high as for MODIS AOD or WRF-Chem $PM_{2.5}$. The HMS smoke product only provides estimates for smoke, which is the primary focus of our search criteria although it also includes phrases related

to general air quality degradation. Additionally, as with MODIS AOD, the HMS smoke product may not be representative of actual surface-level exposure. Finally, the HMS smoke product only provides categorical estimates of "heavy," "moderate," or "light" smoke and likely cannot represent subtle changes in exposure concentration levels as compared to MODIS AOD.

**3.2 Evaluation of All Metrics Compared to Surface Measurements**

While we have shown that our new dataset often correlates well with more-traditional datasets that have been used to estimate smoke and or $PM_{2.5}$ concentrations/exposure, we also investigate whether the percent of Facebook posters can be used to improve estimates when combined with these other datasets. In Figure 4, we compare how good of a predictor each dataset is at estimating $PM_{2.5}$. We show the coefficients of determination for Facebook posters

(4a, similar to 3a but for days where CF < 0.75), MODIS AOD (with CF < 0.75, Figure 4b), the HMS smoke product (Figure 4c), and WRF-Chem $PM_{2.5}$ (Figure 4d) with the surface monitors.

From Figure 4, we can evaluate which dataset best correlates with surface measurements in different regions of the western US.

We summarize these initial findings in Figure 4e, which shows the dataset that was best correlated with the surface measurement at each site (and the $R^2$ had to be greater than 0.5). This figure shows that our Facebook posters dataset is better correlated with actual surface measurements at most sites in our domain for the given time period (5 June – 30 September 2015) compared to other datasets that are typically used to estimate exposure. We do find that MODIS AOD and WRF-Chem $PM_{2.5}$ are better predictors in regions with low populations, such as North Dakota, eastern Montana, and eastern Washington. Additionally, WRF-Chem $PM_{2.5}$ and MODIS AOD are better predictors over much of the eastern US (not shown, $R^2$ values all less than 0.5), which is dominated by anthropogenic emissions during the time period, as these "normal" day-to-day changes in anthropogenic pollution may be less likely to be picked up by our Facebook posting search criteria. To note, we did not optimize the configuration of our WRF-Chem simulation to match surface observations. Changing emissions, meteorology, parameterization choices, grid resolution or time steps may have improved surface-concentration estimates, but the optimal configuration would likely differ by region and time period. However, our results shown in Figure 4 suggest that Facebook posting could be used to help estimate exposure in conjunction with these other datasets.

However, if the aggregate percent of Facebook posters are used to estimate exposure, there may be a few limitations; because, while trends in Facebook posting seem to represent well the variability in surface air quality over our study period at many sites, there is not a simple relationship between posting and $PM_{2.5}$. For one, there did not appear to be a threshold $PM_{2.5}$ concentration at which it was guaranteed that people would start posting, region-wide or at an individual city (e.g. there were cases with high smoke but little posting such as the July event in Fort Collins, CO). There are several potential reasons for this. (1) As noted, on cloudy days, people may not be able to distinguish poor air quality, especially if it is from long-range transport where residents are not aware of a nearby fire. (2) There could be a point of saturation or response fatigue wherein people who have experienced multiple days of smoke may find it less interesting to post about it, or they could have a cognitive bias that causes them to think that air quality has improved in comparison to air quality previously experienced. To test this, we looked also at the time series of the ratio of % of Facebook posters to surface concentrations, and

this ratio does appear to decrease over time during periods of smoke events lasting several days. A decrease throughout the season is only evident at a few sites, although this is difficult to compare because the major smoke event at most sites occurred in late August and early September with few-to-no smoke events occurring afterwards. (3) We noted that occasionally regions with a high Facebook-posting percent was centered over areas where the population had experienced poor air quality on preceding days rather than the current regions of poor air quality. This time shift could suggest that there could sometimes be a lag in either individuals' awareness or in the time it takes to spread information among community-level social networks. Additionally, there could also be persistence in Facebook posts, where air quality might improve in a location but people are still posting about it. Conversely, awareness of events could spread through social network more quickly than an air quality event (such as a smoke plume) is transported such that individuals are discussing an event before it impacts them. Quantitatively, this is difficult to assess as it may be more event related than season-specific. We did compare +/- 1-day lag correlations between Facebook posts and surface measurements for all sites that had daily measurements (as opposed to every third day). Using the same day provided the best correlation at ~90 % of sites. Slightly better correlations were found using the previous day's measurement at several sites in Utah, and using the following day produced better estimates at several sites in Washington and Oregon, where there were broad regions and extended periods of degraded air quality due to local fires.

**3.3 Cloudy Day Modification**

We included the CF criterion for the above analysis for all datasets. We found that filtering out days with high CF improved agreement of Facebook posts and MODIS AOD (Figure 2 and Supplementary Figure 5). This led us to also hypothesize that people may have difficulty distinguishing poor air quality on cloudy days, especially farther downwind of a source. To test this, we also sampled the Facebook posts and surface measurement time series at each site with filtering using the MODIS cloud fraction. Compared to correlations between surface measurements and Facebook posts for the full time period, using only the days with CF < 0.75 improved correlations most noticeably at sites that were generally more than 500 km downwind of fires (such as in Colorado, Wyoming, and Utah, Supplementary Figure 5) but had less impact at sites closer to the 2015 wildfires (Oregon, western Montana, Washington, and Idaho, see Supplementary Figure 1a for fire locations). Cloudiness possibly impacting awareness

on Facebook is seen in the time series for Fort Collins, Colorado in Figure 2a, where, although concentrations were greater during the July event than the August event, the response in Facebook posts was much less. Bellingham, WA was also impacted by smoke during the same period in July, and although lower surface concentrations were measured, the response in Facebook posts was greater. We noted however, that during the July event, the MODIS product reported a cloud cover of 100 % over Fort Collins. For the full time period, filtering out days with a CF > 0.75, improved the $R^2$ between Facebook posts and surface measurements in Fort Collins from 0.33 to 0.54. Alternatively, in Great Falls, MT, which had many nearby fires, filtering only changed the $R^2$ from 0.77 to 0.79, even though roughly the same number of days met the 0.75 criteria for exclusion.

**3.4 Google Trends comparison with Surface Measurements**

We also compared Google trends data to surface measurements of $PM_{2.5}$. Our results are shown in Figure 5 for each search term. As with the Facebook posters, correlations are best in the northwestern US, specifically, Washington, Montana, and Oregon, states that were heavily impacted by smoke in 2015. Although we are comparing to total $PM_{2.5}$, the best correlations were found for not only "air quality", but also "wildfire" and "smoke", which, as with the Facebook posters, we might expect since wildfire smoke was the source of the most variability in surface $PM_{2.5}$ during this time period. The search terms that are more related to urban pollution ("pollution", "smog", "haze" and "ozone") have much lower correlations, and sites that do have $R^2>0.1$ are generally in urban areas or far downwind of smoke. "Ozone" in particular was not well-correlated with $PM_{2.5}$ measurements (all $R^2 < 0.22$), which should be expected since ozone concentrations and $PM_{2.5}$ concentrations are not always well-correlated (e.g. Reisen et al., 2011).

**3.5 Google Trends search term comparison**

We also used the Google Trends data to analyze our Facebook search term criteria because we were not able to do this within the Facebook posters dataset. We chose several words that might be associated with "air quality" and determined the correlations between each word for each DMA as shown in Figure 8. As with the actual concentrations of $PM_{2.5}$, we find that "air quality" is, in general, more associated with "smoke" and "wildfire" than words more commonly associated with urban sources like "smog", "haze", "pollution", and "ozone". In Sachdeva et al. (2016), the authors found that distance from the fires impacted the content of postings about the fire, and we also note some differences in our correlation maps based on distance. For example,

closer to the fires (WA, OR, ID, MT), "air quality" is more associated with "smoke", while farther away (CO, NV, UT, WY), "air quality" is more associated with "wildfire". At these sites, "air quality" is also better correlated with "wildfire" than "smoke", which may suggest that people are aware of the impact of the wildfires on air quality, but not able to see smoke.

However, Google Trends are scaled by popularity in each region and data on only very popular terms are available. This could lead to a discrepancy in that the same amount of people may be searching for these terms in different regions, but the relative popularity may be very different compared to other search terms, especially if there are other physical sources of "smoke" or "air quality" in a region. "Ozone", "smog", and "pollution" (terms that may be more associated with

urban air pollution), are not well-correlated with "air quality", "smoke", or "wildfires" over our study period; however, "haze" is moderately correlated in WA, OR, and CO (Figure 6).

**3.6 Geographically Weighted Regression Test Case for Washington state**

As a first case test to evaluate the usefulness of this Facebook posters dataset in a statistical model, we compared two geographically weighted regression model estimates using

MODIS AOD and WRF-Chem $PM_{2.5}$ with and without the Facebook posters. From Figure 4, we see that WRF-Chem $PM_{2.5}$, MODIS AOD, and this Facebook posters dataset are all correlated with surface $PM_{2.5}$ in Washington state, and the best correlated variable varies between surface sites. Therefore, a regression model could allow us to leverage the strengths from each dataset to create an improved estimate.

In Figure 7, we show the results for our regression models with and without the Facebook posts. We see that including the Facebook posts in the regression model leads to improved $R^2$ values at many of the sites in Washington (only one site shows a decrease, Figure 7e). Additionally, for the full dataset (of all sites and all days), there is an improved $R^2$ (0.66 compared to 0.58), slope (0.60 compared to 0.52), and a smaller error. While, these

improvements may be small; we find this is in part because the Facebook posts explains much of the same variability as WRF-Chem $PM_{2.5}$ (and better explains variability in the urban region around Seattle, WA). We also did not account for cloudy days in our regression analysis. Including information on cloud cover could potentially improve our regression model, which will be investigated further in ongoing work on this analysis.

**4   Conclusions**

In this paper, we introduced a novel concept of using de-identified, aggregated counts of Facebook posts mentioning smoke, haze, or air quality to determine exposure by comparing to traditional datasets and in a regression model. We also looked at Google Trends data for the same time period and compared it to surface observations. The Facebook posts were useful in regions meeting two conditions: (1) the region was impacted by LFS, and (2) there was a large-enough population posting to Facebook. The Google Trends data were also best correlated in regions impacted by smoke, however, it is aggregated at a much coarser resolution (DMA-level), therefore the impact of population density is unclear. For regions that meet these two criteria, the Facebook posts agreed well with more-traditional datasets routinely used for estimating smoke concentrations. In fact, the dataset was often a better predictor of surface $PM_{2.5}$ than several of these other methods and/or datasets (MODIS AOD, HMS smoke product, WRF-Chem $PM_{2.5}$). Therefore, this Faceboook posters dataset could be useful in determining spatial extent of exposure between surface monitors.

In further investigating regions and time periods of poor agreement, we noted that the cloud cover negatively impacted our correlations, suggesting that some environmental factors might impact people's awareness. We also found that in some regions, correlation improved when comparing to the previous or following day, possibly suggesting some influence of social media on awareness. Some of the disagreement could also be due to our search criteria, which could be further refined to reduce the number of false negatives (not recognizing a post is about air quality) and false positives (including posts that are not about air quality) that likely occur with colloquial conversations. Other studies, which have relied on Twitter messages, have been able to optimize this process by examining subsets of individual posts ("Tweets") to test for false positives. However, again, because this dataset does not provide information on individual posts, this is difficult to do solely within this dataset, but we do plan to test different search criteria in the future to aid in optimizing our dataset.

Even with some of these limitations, we demonstrated that this percent of Facebook posters dataset has strong potential to be used to estimate exposure to poor air quality. Sachdeva et al. (2016) has shown similar results with Twitter data, but only for a single fire in California. We believe that Facebook posts could provide some specific advantages over Twitter. Facebook is the most widely used social-media site in the US, with 70 % of its participants active daily (Duggan et al., 2015), compared to 36 % for Twitter. Additionally, only 1 % of Twitter posts are

geo-referenced (Thom et al., 2013), and Google Trends relies on a subset of searches for a large region. In Sachdeva et al. (2016), the actual analysis only included 1297 tweets from a 45-day period covering a region of 40,000 km$^2$ in California and Nevada, and their statistical model was built from 705 tweets for a 37-day period covering a 7,500 km$^2$ area. With a broader user-base,

Facebook posts could potentially provide better spatial resolution over a broader region. Therefore, this dataset of de-identified, aggregated counts of posts, could be very useful for estimating population-level exposure. While we showed that Google Trends data were also moderately well-correlated with surface PM$_{2.5}$ in the Northwest, results were only available for DMAs, of which there are only 210 in the US, leading to significantly less spatial information in

the Google Trends data than with our percent of Facebook posters (which has results for >20,000 cities in the US). In 2015, there was a broad region of smoke over much of the US; therefore, correlations with Google Trends may be much higher than if we compared to years with only localized smoke events. Finally, we presented a first test case using the percent of Facebook posters in a statistical model to predict surface concentrations in Washington state for June –

September 2015, showing improvements in slope and R$^2$ values and a reduced error in predicted PM$_{2.5}$. We plan to extend this work in order to provide improved estimates of smoke exposure for the whole western US for the 2015 summer, which will then be used to quantify the health responses associated with exposure to wildfire smoke. Improving the understanding of these specific health effects can potentially aid the public and decision-makers on when and how to

take measures to reduce exposure. While social media will not be able to completely replace traditional methods of estimating exposure, social media datasets could currently improve estimates without the costly investment of additional surface monitors. Using social media datasets as a proxy for exposure, also lends itself to analysis of people's response and understanding of smoke exposure (Sachdeva et al., 2016), which cannot be measured by

traditional exposure methods.

## 5    Data Availability

The 24-hour average concentrations of total PM$_{2.5}$ mass are available from the EPA Air Quality System at epa.gov/aqs, and the IMPROVE PM$_{2.5}$ data are also available at

views.cira.colostate.edu/fed/. The Collection 6, MODIS Level 2 10-km AOD products from the Terra and Aqua platforms are available at ladsweb.nascom.nasa.gov. The HMS fire and smoke

analysis product is available through satepsanone.nesdis.noaa.gov. Google trends data are available at google.com/trends. Our WRF-Chem model output (daily, 24-hour average surface concentrations) is available at http://hdl.handle.net/10217/177042. The Facebook posts data retrieval was conducted internally at Facebook by a Facebook data scientist. To preserve the

privacy of Facebook users and in accordance with the data use agreement, we are unable to provide the Facebook posters data. However, we do provide daily maps of the raw and gridded aggregate percent of Facebook posters at http://hdl.handle.net/10217/177043.

## Acknowledgements

This work was funded by NASA Applied Science grant NNX15AG35G.

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

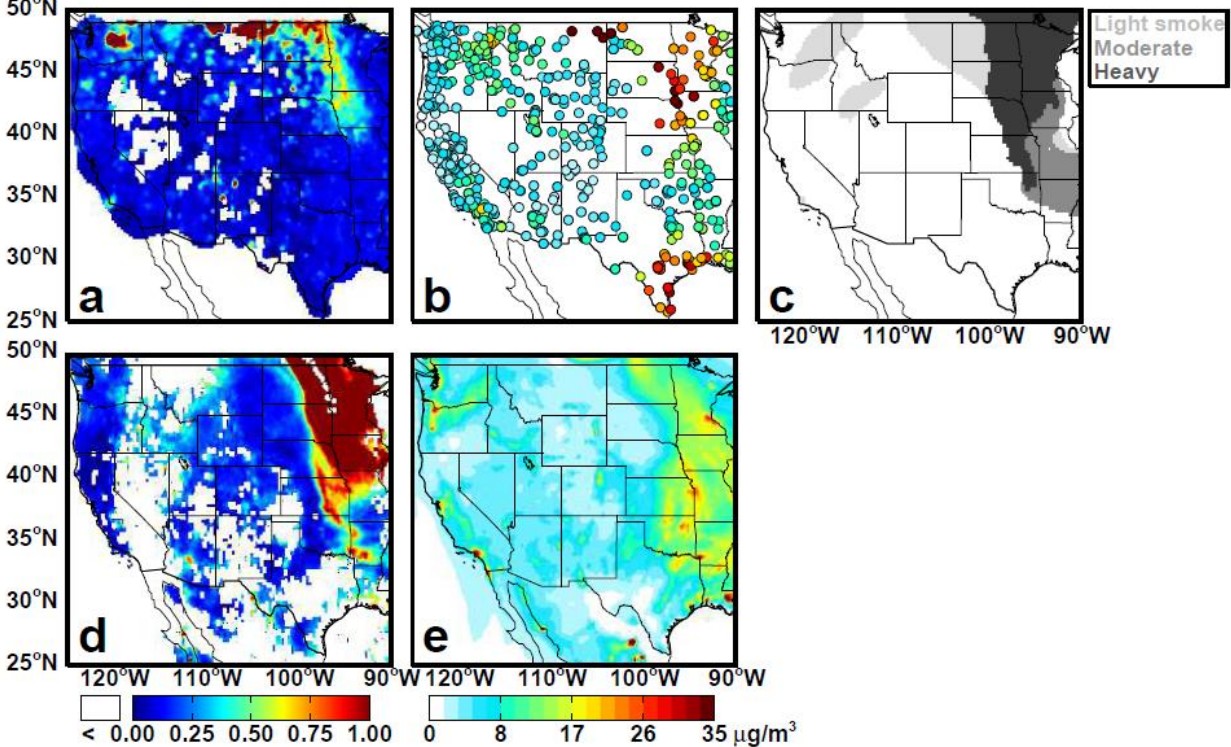

**Figure 1.** Example of datasets for 29 June 2015. a.) Population-weighted (Equation 1) percent of Facebook posters meeting criterion (white signifies regions with weighted population < 10), b.) 24-hr average surface PM$_{2.5}$ concentrations from surface measurement sites, c.) gridded HMS smoke product, d.) gridded, unfiltered MODIS-Aqua and MODIS-Terra AOD (white signifies no valid observation), and e.) WRF-Chem simulated 24-hr average surface PM$_{2.5}$ concentrations.

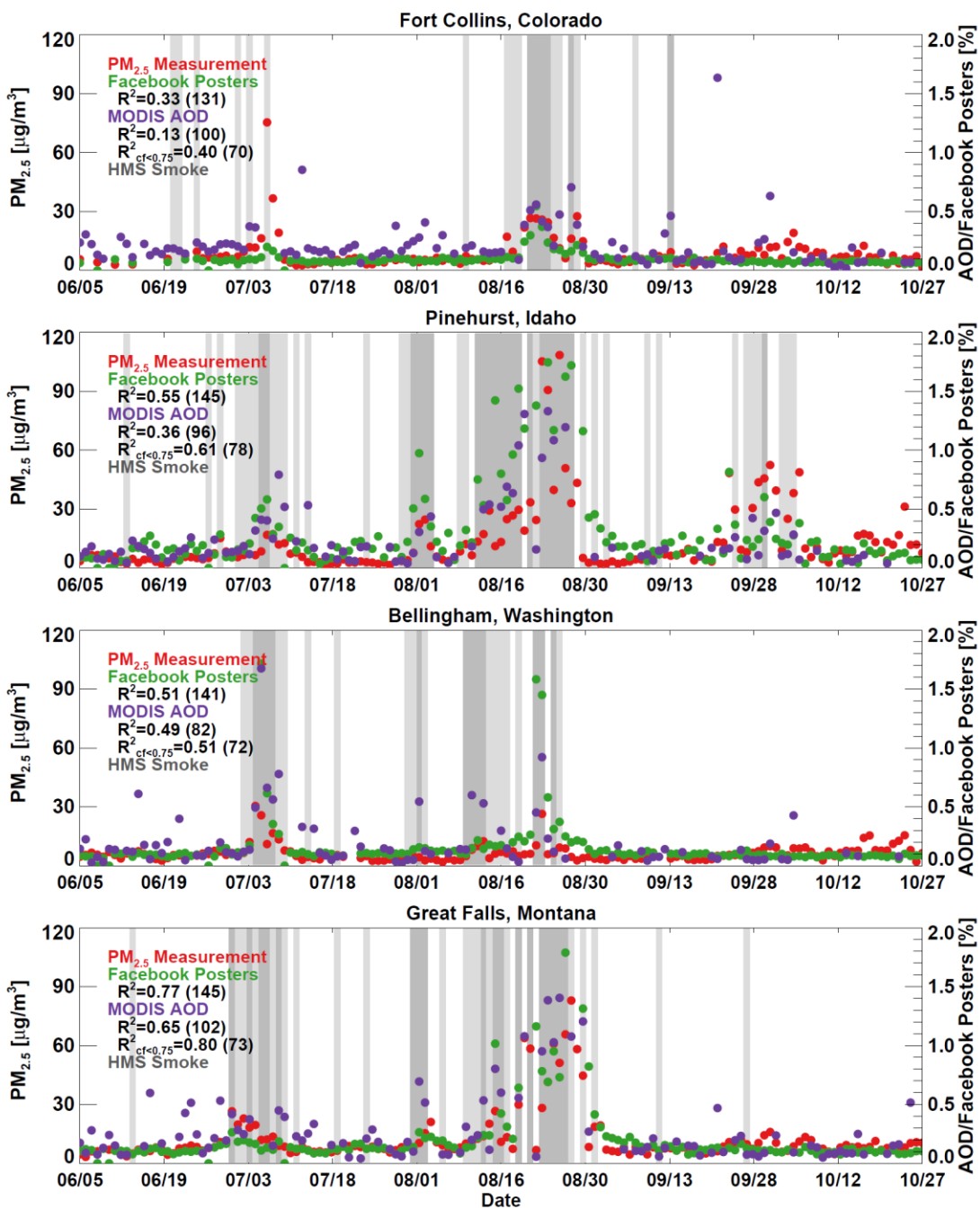

**Figure 2.** Time series of measured surface PM$_{2.5}$ concentrations (red), gridded and population-weighted percent of Facebook posters (green), MODIS AOD (purple), and days with HMS-denoted light (light gray) and moderate/thick (dark gray) smoke at (a) Fort Collins, CO; (b) Pinehurst, ID; (c) Bellingham, WA; and (d) Great Falls, MT for 5 June – 27 October 2015. R$^2$ values for each dataset with the surface measurement are given along with the number of days available for the calculation noted in parentheses.

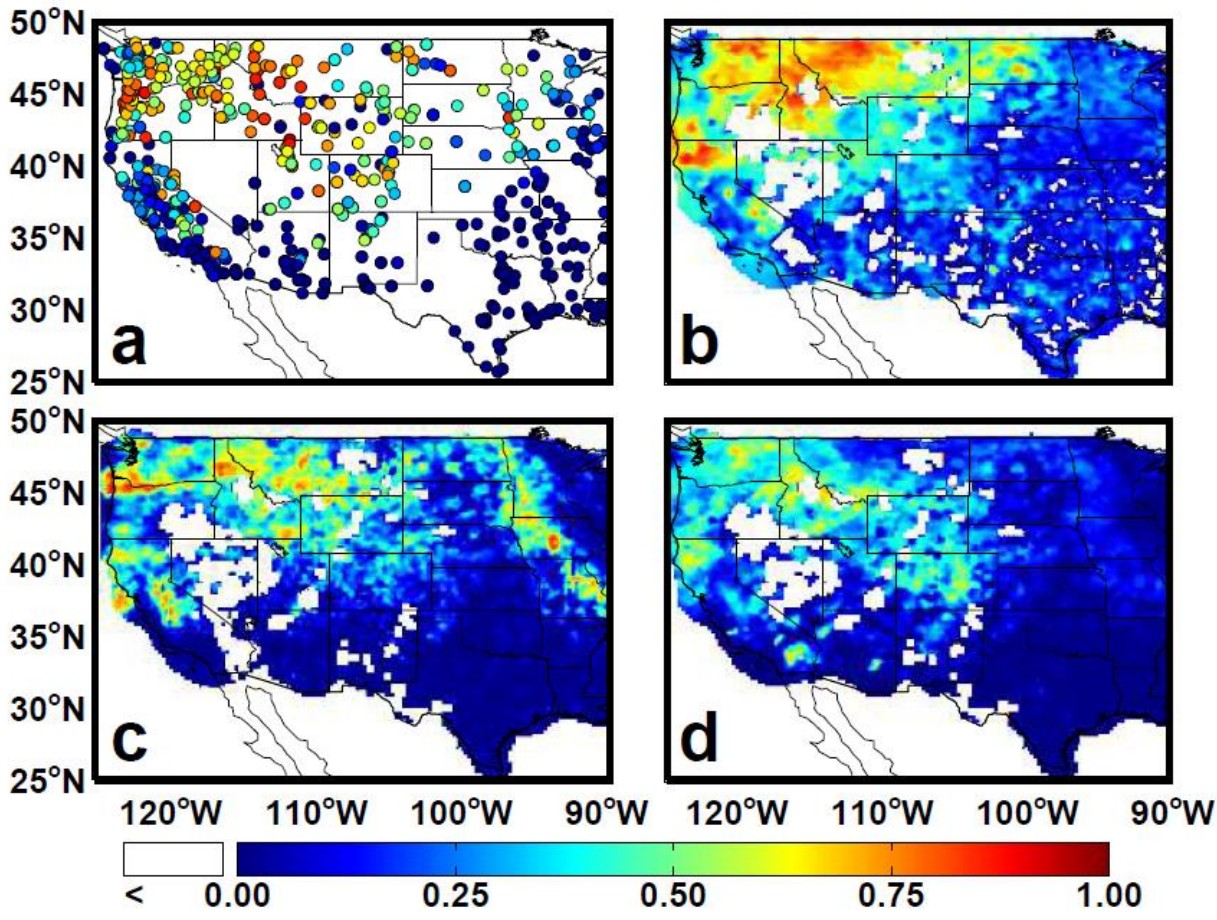

**Figure 3.** $R^2$ values for % Facebook posters and a.) IMPROVE and EPA-AQS surface measurements of $PM_{2.5}$ (for sites with > 35 days of measurements), b.) WRF-Chem $PM_{2.5}$, c.) MODIS AOD when cloud fraction was below 0.75 and d.) HMS smoke product for the period of 5 June – 30 September 2015.

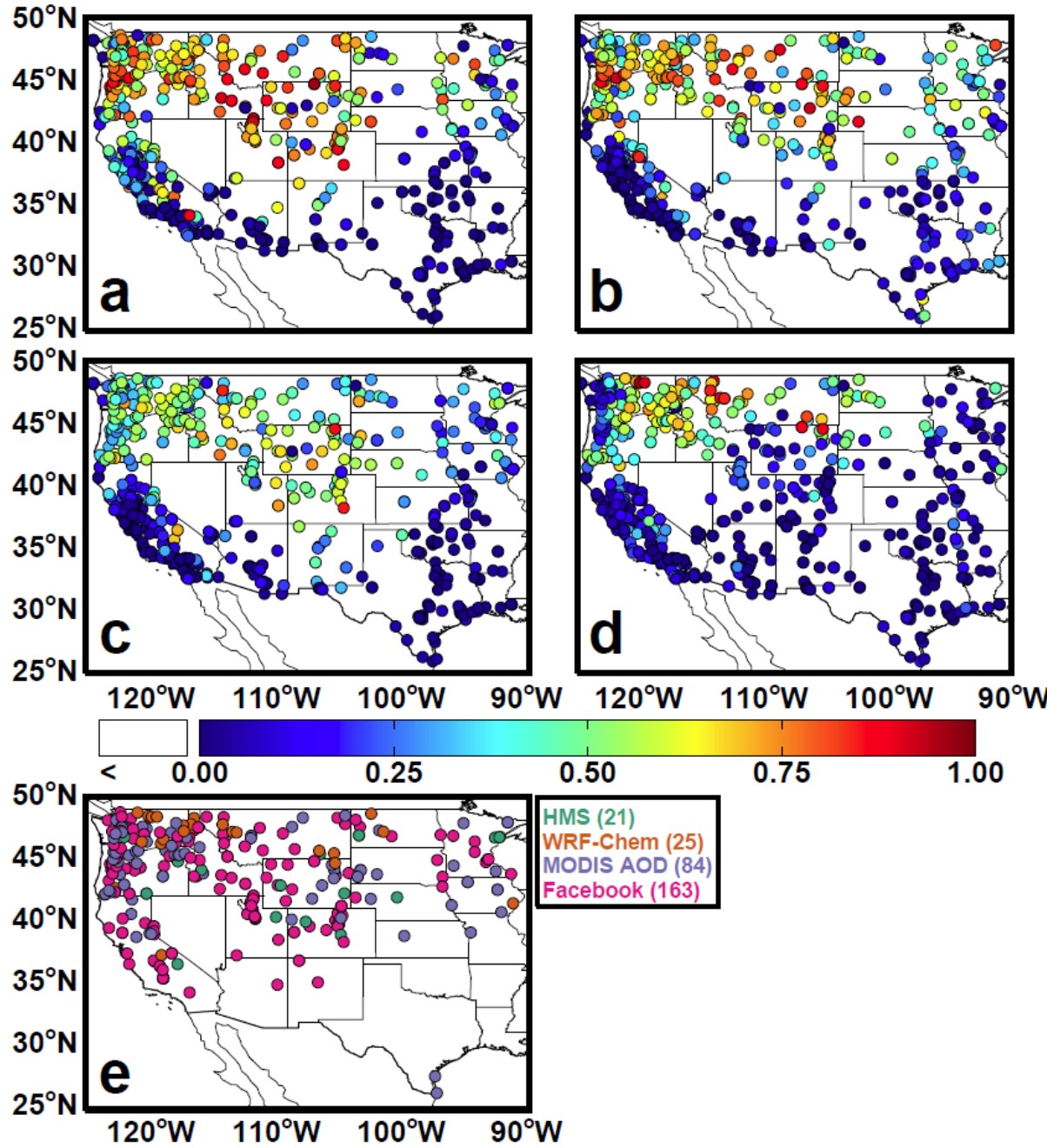

**Figure 4.** $R^2$ values for surface measurements of $PM_{2.5}$ with a.) percent of Facebook posters (CF<0.75), b.) MODIS AOD (CF<0.75), c.) HMS smoke, and d.) WRF-Chem simulated $PM_{2.5}$, for the period of 5 June – 30 September 2015. e.) Product (HMS Smoke, WRF-Chem $PM_{2.5}$, MODIS AOD, or Facebook posters) that has the highest $R^2$ compared to surface measurements for the time period of 5 June – 30 September 2015 (sites are shown only if the resulting $R^2$ > 0.5). Number of sites in western US (domain shown) where product has highest $R^2$ (and $R^2$ > 0.5) is given in parentheses.

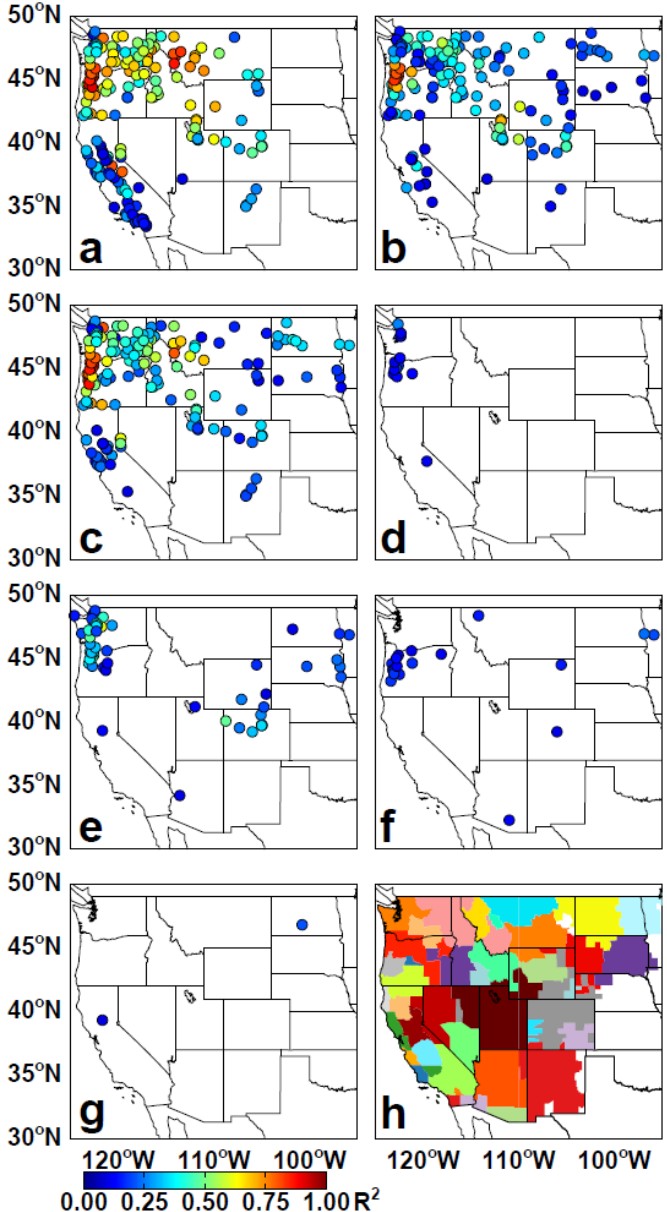

**Figure 5.** $R^2$ values at each measurement site for surface measurement and Google Trend search trend (a) "air quality", (b) "wildfire", (c) "smoke", (d) "pollution", (e) "haze", (f) "smog" and (g) "ozone." Only sites where $R^2 > 0.1$ are shown. The 48 DMAs considered are shown in (h).

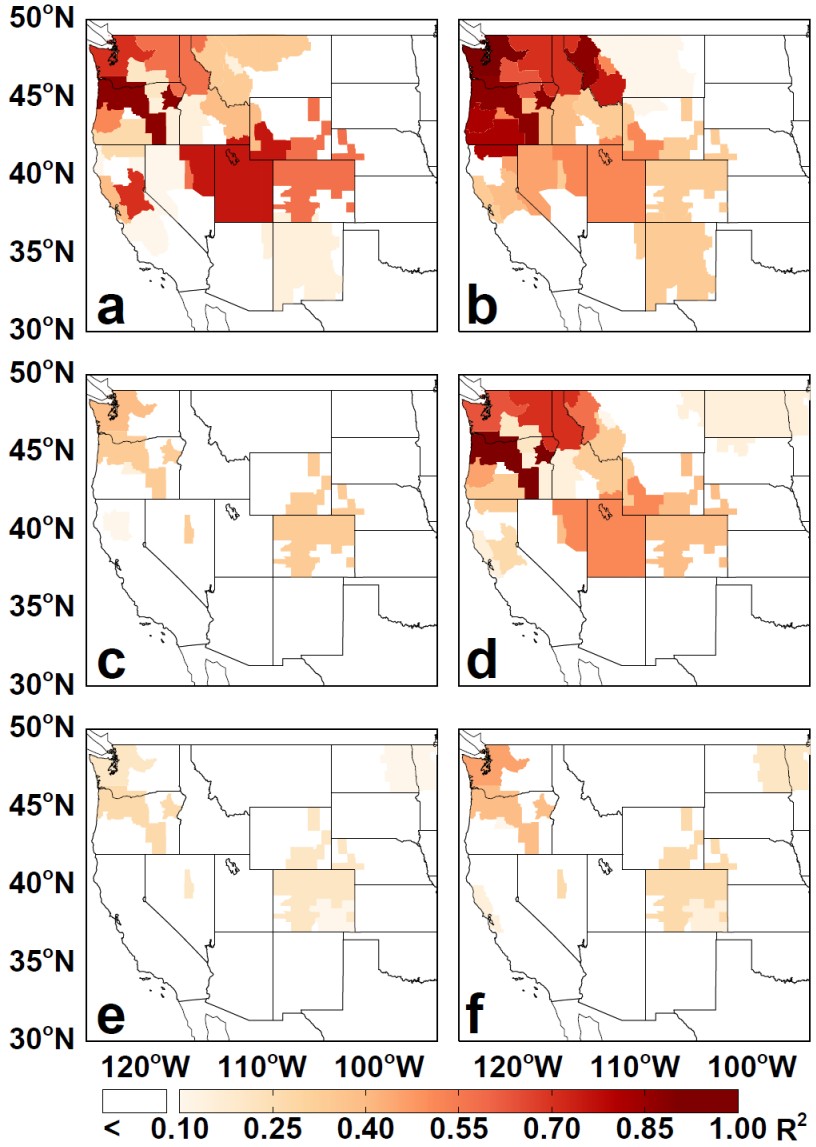

**Figure 6**. $R^2$ values for pairs of Google Trends search terms (a) "air quality" and "wildfire", (b) "air quality" and "smoke" (c) "air quality" and "haze", (d) "wildfire" and "smoke", (e) "wildfire" and "haze" and (f) "smoke" and "haze" for June –September 2015.

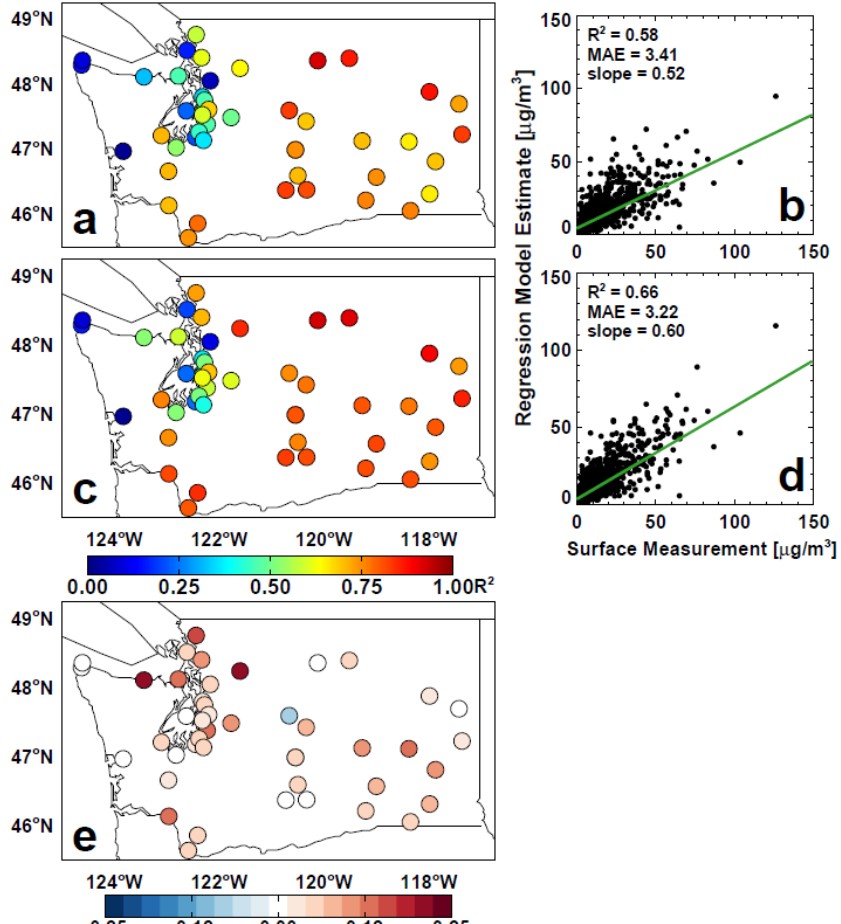

**Figure 7.** $R^2$ values at each measurement site for surface $PM_{2.5}$ and regression model estimate (a) using MODIS AOD and WRF-Chem $PM_{2.5}$ and (c) using MODIS AOD, WRF-Chem $PM_{2.5}$, and percent of Facebook posters for 5 June – 30 September, 2015 and the difference in $R^2$ between the two regression models (with Facebook posters- without Facebook posters). Also, scatterplots for all daily measured $PM_{2.5}$ and corresponding regression model estimates in the domain (b) using MODIS AOD and WRF-Chem $PM_{2.5}$ and (d) using MODIS AOD, WRF-Chem $PM_{2.5}$, and Facebook posters.