# Peer review of "Status Update: Is smoke on your mind? Using social media to assess smoke exposure"

_Atmospheric Chemistry and Physics, 2017_

## Author Comment (AC1) · 20 Jan 2017

There is an error regarding the format of the references in several places within the manuscript. All references are included in the list of references in the manuscript, but the reference manager format within the paper was inconsistent.

The numbered references should be as follows:

On Page 3, line 14: (e.g. 3) should be (e.g. Pope III et al., 2009)

On Page 3, line 23: (e.g. 14, 15) should be (e.g. Henderson et al., 2011; Yao et al., 2013)

On Page 3, line 23: (e.g. 16-19) should be (e.g. Brauer et al., 2015; van Donkelaar et al., 2015; Reid et al., 2015; Yao and Henderson, 2013)

[Figure]

On Page 4, line 13: (e.g. 14-16) should be (e.g. Broniatowski et al., 2013; Crooks et al., 2013; Ginsberg et al., 2009)

On Page 16, line 10: (e.g. 42) should be (Sachdeva et al., 2016)
* * *
**ACPD**

---

## Referee Comment (RC1) · S. Henderson (Referee) · 17 Feb 2017

I really don't understand the process of review for this journal. I'm not sure whether my review is going to be posted as a comment on the manuscript, and whether that should influence the way I write the review? Regardless, I am going to use the same type of format I always use. Please see supplemental PDF for formatted version

Summary:

The exposure of interest is smoke from landscape fires. The authors have compared data from social media (Facebook) and online searches (Google) with data from more conventional methods used to assess smoke exposures: PM2.5 measurements from two sources (AQS and IMPROVE), AOD measurements from MODIS, integrated satellite plume footprints from HMS, and PM2.5 estimates from WRF-Chem. They have

also compared the AOD, HMS and WRF-Chem method with the PM2.5 measurements which are the de-facto gold standard. They report that that Facebook posts are useful for assessing population exposure, especially when the data are population-weighted. Their correlation with measured PM2.5 is comparable with correlations observed for the other metrics. When combined with the other metrics they can improve models fitted to the PM2.5 measurements. The Google search data were largely used to compare the utility of different keywords when assessing smoke exposures.

First and foremost, I am impressed by this work. I think it is a very nice and thorough contribution to the literature. My concerns are not about the quality of the science, but about the clarity of its presentation. There are A LOT of ideas in this manuscript, and I think that more careful consideration of its structure would improve its accessibility to readers. Major and minor concerns are listed below.

Major concerns:

- More information about the Facebook data is needed. My interpretation is that the study team provided a list of search terms to Facebook, and was provided with a daily percentage value for each community. The team never saw the posts (this is made clear) and the results for all search terms were lumped together (this is less clear), making it impossible to disaggregate "smoke" from "haze", for example. More specific detail is needed. It was also not clear whether the values reflected the proportional of posts or the proportion of individuals posting. If one noisy person was posting about smoke in a small community, this could make a difference. Either is acceptable, just be clear about what is measured and its limitations.

- If my interpretation is correct, the primary utility of the Google analyses is the ability to disaggregate results for different keywords and to evaluate which are most strongly correlated with smoke. I recommend reframing methods section on the Google data to address this one clear objective.

- On a related note, the authors used a limited number of keywords in their search,

and I think this deserves comment in the discussion – especially with respect for the potential of machine learning to help refine keywords in future work. In our more limited work on this topic we found many posts that make statements such as "Smells like a campfire out here!" and "Where's the fire?" which would not be captured by the methods described.

- The value of presenting the raw and weighted Facebook posts is not clear given that the weighting seems necessary. Would highly recommend removing this one complexity from the other many complexities. Simply state that the data were weighted and describe the methods used to weight them.

- Suggest the authors focus more specifically on smoke rather than more generally on air quality, as smoke is really the thing they are trying to capture. A statement in the introduction about the fact that air quality can degrade for other reasons would be useful, especially if analysis on the Google searches is reframed, but then they could stick to the idea of smoke throughout. Rather than saying "exposure to degraded air quality" just say "smoke". It's a lot simpler, and it's what the manuscript is about.

- Similarly, they could simply define "landscape fire smoke" (currently wildland fire smoke, which omits the agricultural category – often important in the US) in the introduction and then use LFS throughout

- One of the most challenging things about the current manuscript is that different phrases often get used for the same concepts. I encourage the authors to give each smoke exposure metric a single name and to rigorously use that name consistently throughout. For example: Facebook posts; Google searches; PM2.5 measurements (specifying AQS or IMPROVE where necessary); MODIS AOD; HMS plumes; and WRF-Chem PM2.5. Further, suggest that all of these metrics go under single Smoke Exposure Metrics subheading in the Methods section, and that each gets its own paragraph with its name in bold at the beginning. This will more easily help readers to refer back to the methods while they are pondering the results. Also, be clear up front that

analyses were done at the 24-hour time scale.

- Methods section currently gives no information about how Facebook posts were compared with other metrics (temporal correlations, spatial correlations, etc). Much of this information is erroneously included in the Results. Suggest two subheadings be added to Methods: (1) Comparison of Facebook Posts with Conventional Metrics, and (2) Comparison of Other Metrics with PM2.5 Measurements. Main conclusions are that (1) Facebook Posts are correlated with conventional metrics, and (2) they are as correlated with the gold standard as other metrics. As such, suggest they clearly frame the Methods so that the conclusions naturally follow.

- Overall, suggest the following subheadings for Methods: Smoke Exposure Metrics; Comparison of Facebook Posts with Conventional Metrics; Comparison of Other Metrics with PM2.5 Measurements; Assessing the Effects of Cloudy Days; Regression Case Study in Washington State; and Using Google Searches to Evaluate Keyword Utility.

- As such, suggest the following subheadings for Results and Discussion: Facebook Posts Compared with Conventional Metrics; Other Metrics compared with PM2.5 Measurements; Cloudy Day Modification; Regression Case Study; and Keyword Utility for Smoke Detection. It seems like there should be at least one table allowing readers to compare correlations between the main metrics (Google searches omitted).

- Were cloudy days controlled for in the regression analysis? Sounds like they should be, given the findings of that sub analysis.

- Finally (and I know this is long – I'm sorry!), a good paper should stand alone without its figures and tables just as the figures and tables should stand alone without the paper. The authors most often use statements such as "Agreement between MODIS AOD and Facebook posts are shown in Figure 3" where they should use "Agreement between MODIS AOD and Facebook posts was moderate (Figure 3)". Similarly, "In Figure 2 we also show example time series of Facebook posts" should be "An example

of the time series of Facebook posts and other metrics shows that. . .(Figure 2)".

Minor concerns:

- The statement about cloud cover in the abstract is not put in enough context to make sense. For example, one would assume that Facebook compares poorly with AOD on cloudy days because AOD performs poorly on cloudy days – not because Facebook users (who have noses) perform poorly on cloudy days. The authors do present intriguing evidence to the contrary in the results, but this statement in the abstract shakes their credibility.

- There's a lot of weird and insistent use of hyphens in the text. For example air-quality is not conventionally hyphenated. Please review carefully and correct for common usage.

- Paragraph numbed 15-25 on Page 3 has some really long and complex sentences. Please break up for more clarity.

- Referencing still using numbers rather than names in some places (line 23 on Page 3 and others).

- There's a lot of use of the word "determine" which is quite strong. Its definition implies exactitude. Suggest words like "assess" and "evaluate" are more appropriate. Recommond title be changed to ". . . : Using social media to assess population smoke exposure".

- AOD and MODIS used before their definitions.

- Page 4: what are "air quality exposure" and "risk exposure assessment"? Do they mean "air pollution exposure" and "exposure assessment"?

- Social media = plural, treated as singular

- Data = plural, treated as singular

- Methods for population weighting / gridded estimates on Page 5 not very clearly described and nor is the rationale. Suggest statement about why this needs to be done and then described as a weighted spatial interpolation (which I think it is).

- US or U.S. – choose one.

- Page 12 brings up the question of visibility, for which the US has good data. Do not suggest that the authors do further analyses, but do suggest that they give thought to what such analyses could help to elucidate.

- Pages 12 and 13, both line 15: because, not since

- Figure 1: Suggest showing HMS plumes and PM2.5 Measurements in separate plots, just to make it really clear that we are dealing with five exposure metrics.

- Figure 2 is. . .er. . .a lot. Reducing to population-weighted only Facebook posts would help. Do you really need cloudy days on here? Can you do it with something other than dogs, if so? Why these four locations? Should be described in Methods.

- Time series is two words

- Things that are alike are compared with each other, not to each other

Please also note the supplement to this comment:
http://www.atmos-chem-phys-discuss.net/acp-2017-26/acp-2017-26-RC1-supplement.pdf

─────────────────────

---

## Referee Comment (RC2) · S. Henderson (Referee) · 17 Feb 2017

Good grammar is very important to me, so I am sympathetic to your crusade. Thanks for the explanation. However, I do feel this nuance is going to be lost on most readers as it was lost on me. Maybe a foot note about it on first application?
* * *

---

## Short Comment (SC1) · 17 Feb 2017

Thanks, Sarah for your thoughtful review of our manuscript. While Bonne works on the full response, I wanted to respond about your hyphen comment because proper hyphenation of compound adjectives (http://www.grammarbook.com/punctuation/hyphens.asp) is a (strange?) personal crusade of mine.

I like to use the following sentence as an example...

"We need more accurate measurements."

Do we need measurements that are more accurate? Or do we already have measurements that are accurate enough, but rather we need more of these accurate measure-

ments. The former should be "more-accurate measurements", and the latter should be "more accurate measurements". However, when an author does not establish that they always hyphenate compound adjectives, even when the meaning is likely obvious from context, sentences like this (without a hyphen) are ambiguous.

Thus, something like "air quality data" seems obvious and the use of "air-quality data" seems unnecessary (we clearly don't mean "quality data" of the air variety, which is what the strict meaning of "air quality data" without a hyphen is). However, by using the hyphen, we are establishing that if we have a compound-adjective situation, we will always use a hyphen. This way, if we say, "We need more accurate measurements." you can be confident that we already have the means of making accurate measurements, we just need more of these measurements. If we had meant the other definition, we would have used a hyphen.

Note that we do not use a hyphen for "air quality" when it is not an adjective, "...improvements in air quality." (if we did, this was a mistake, and we will check for this). We only hyphenate when it being used as a compound adjective.

---

## Short Comment (SC2) · 17 Feb 2017

Yeah, that's not a bad idea... might spread the gospel more quickly too.
* * *

---

## Editor Comment (EC1) · D. Topping (Editor) · 24 Apr 2017

In this paper the authors explore the power of social media data to improve data coverage on smoke exposure. As the need for increased data density in atmospheric exposure generally progresses, it is highly likely that more studies will rely on the 'citizen sensors' approach. I find the study a refreshing addition to the often stagnant observation based literature. It adds to the already wealthy cross-disciplinary arm of ACP and I enjoyed reading it.

I would like to see this published in ACP after some comments are addressed below.

Could you remove multiple contributions from a particular individual from Facebook? How do you remove a particular biased commentary from a subset of users? I was interested in the lack of threshold PM2.5 concentration for people to start posting. Could

[Figure]

this be a factor? If a small number of users are relying on available monitoring data, then reporting this, they might be driving a wider response. This isn't necessarily a negative feature, of course, but has parallels in social media coverage of viral outbreaks.

What percentage of facebook users are you actually obtaining? For example, twitter restricts access to a small percentage unless a fee is paid. Could you add this information to the manuscript?

I often wonder how much an individual response is due to reporting on a news item/political debate rather than commentary on conditions experienced at any point in time. As with some practices in sentiment analysis, it might be useful to analyse bigrams/trigrams for a given post. Is that data available?

I appreciate the difficulty in providing social media data, having been personally rejected from other journals on this commonly known technicality. Would it be possible to provide a little more detail on the process of Facebook data retrieval for those who might want to replicate a similar study at least?

Regarding the regression model, was there a particular reason to opt for linear combinations of predictor variables? I wonder if the accuracy of your technique might be increased by even a simple decision tree, or ensemble method, an additional variables. Using k-folds cross validation and variable selection this might generate a more widely applicable method.

A minor comment on the line: 'social media datasets could currently improve estimates without the costly investment of computer modeling.' I would add this really depends on the application. If you were to fit a multivariate regression model to actual post content, with access to many hundreds of thousands of posts, the time to train a model varies with amount of data used. I leave it to the authors to decide on whether to retain this.

---

## Author Comment (AC2) · 8 May 2017

**The authors thank the two reviewers for their helpful comments. Responses are in bold below each specific comment in italics.**

Reviewer 1:
*Summary:*
*The exposure of interest is smoke from landscape fires. The authors have compared data from social media (Facebook) and online searches (Google) with data from more conventional methods used to assess smoke exposures: PM2.5 measurements from two sources (AQS and IMPROVE), AOD measurements from MODIS, integrated satellite plume footprints from HMS, and PM2.5 estimates from WRF-Chem. They have also compared the AOD, HMS and WRF-Chem method with the PM2.5 measurements which are the de-facto gold standard. They report that that Facebook posts are useful for assessing population exposure, especially when the data are population-weighted. Their correlation with measured PM2.5 is comparable with correlations observed for the other metrics. When combined with the other metrics they can improve models fitted to the PM2.5 measurements. The Google search data were largely used to compare the utility of different keywords when assessing smoke exposures.*

*First and foremost, I am impressed by this work. I think it is a very nice and thorough contribution to the literature. My concerns are not about the quality of the science, but about the clarity of its presentation. There are A LOT of ideas in this manuscript, and I think that more careful consideration of its structure would improve its accessibility to readers. Major and minor concerns are listed below.*

**Thank you, Dr. Henderson, for your review and your positive comments on the paper! Specific responses are noted in bold below the italicized reviewer comment.**

*Major concerns:*
*- More information about the Facebook data is needed. My interpretation is that the study team provided a list of search terms to Facebook, and was provided with a daily percentage value for each community. The team never saw the posts (this is made clear) and the results for all search terms were lumped together (this is less clear), making it impossible to disaggregate "smoke" from "haze", for example. More specific detail is needed. It was also not clear whether the values reflected the proportional of posts or the proportion of individuals posting. If one noisy person was posting about smoke in a small community, this could make a difference. Either is acceptable, just be clear about what is measured and its limitations.*
**Yes, results for all the search terms were lumped together, so we were not able to disaggregate "smoke" from "haze," which is why we were careful to not assume that the signal in the Facebook data was all from "smoke." As stated in the text "Our dataset is the percentage of Facebook posters" (i.e. individuals). To note, for privacy reasons we do not know the absolute number of posters on each day. However, as with the Google Trends data, the search does not include when the same individual is posting (or searching in the case of Google Trends) multiple times a day about smoke. We added a few clarifications to the text.**

**The paragraph now reads: Our dataset is the percentage of individual Facebook posters in each US city that used any of the following words: "smoke", "smoky", "smokey", "haze", "hazey", or "air quality" in a post, while attempting to filter out reference to cigarette smoking and other phrases not related to air quality (see Supplement). The search generates de-identified and aggregated data of all posts and removes double counts (i.e. when an individual posts multiple times a day); no individual's text was viewed by researchers. Our goal was to focus on wildfire smoke because wildfire smoke often leads to extreme air quality degradation over broad regions of the US in the summertime. However, because this list includes "air quality" and "haze" (and results were all aggregated), this search criterion could also highlight trends in Facebook posts discussing air quality degradation due to other emissions, such as fossil-fuel combustion. Geographic location at the city level is determined by IP address. Data were provided for 5 June through 27 October 2015.**

*- If my interpretation is correct, the primary utility of the Google analyses is the ability to disaggregate results for different keywords and to evaluate which are most strongly correlated with smoke. I recommend reframing methods section on the Google data to address this one clear objective.*

**You are correct that this is the primary utility. However, we also wanted to show that our Facebook dataset is potentially more useful than other internet-behavior datasets. We have added this sentence to the text: "Our reason for including this analysis is twofold: (1) to compare the results of our Facebook comparison to results using another internet behavior dataset and (2) to determine which keywords are most strongly correlated with PM$_{2.5}$ (as our Facebook dataset is an aggregated result for all search terms)."**

*- On a related note, the authors used a limited number of keywords in their search, and I think this deserves comment in the discussion – especially with respect for the potential of machine learning to help refine keywords in future work. In our more limited work on this topic we found many posts that make statements such as "Smells like a campfire out here!" and "Where's the fire?" which would not be captured by the methods describe.*

**We agree and have added this to the discussion by editing the following sentence: "Some of the disagreement could also be due to our search criteria, which could be further refined to reduce the number of false negatives (not recognizing a post is about air quality) and false positives (including posts that are not about air quality) that likely occur with colloquial conversations."**

*- The value of presenting the raw and weighted Facebook posts is not clear given that the weighting seems necessary. Would highly recommend removing this one complexity from the other many complexities. Simply state that the data were weighted and describe the methods used to weight them.*

**We assume that Dr. Henderson is referring to Figure 2. We presented both to show (1) that in regions with low populations (Pinehurst, ID as the example), population weighting improves the agreement and (2) in regions with larger populations (Fort Collins, Bellingham, and Great Falls) population weighting does not impact the results. However, to improve Figure 2, we moved this comparison to the Supplement.**

*- Suggest the authors focus more specifically on smoke rather than more generally on air quality, as smoke is really the thing they are trying to capture. A statement in the introduction about the fact that air quality can degrade for other reasons would be useful, especially if analysis on the Google searches is reframed, but then they could stick to the idea of smoke throughout. Rather than saying "degraded air quality" just say "smoky". It's a lot simpler, and it's what the manuscript is about.*

**Because the Facebook search term results are lumped together, it is not possible to disaggregate when people are posting about smoke vs other sources of poor air quality. We have added the following sentence to the introduction: "While there can be many different sources of poor air quality, the highest PM$_{2.5}$ concentrations measured during the study period were in regions and during time periods associated with wildfire smoke." While this manuscript is mainly focusing on smoke (because that is when the agreement is best), our methods here cannot say that the only source of degraded air quality in the time period is smoke. While we lose some simplicity, we would like the text to keep this distinction.**

*- Similarly, they could simply define "landscape fire smoke" (currently wildland fire smoke, which omits the agricultural category – often important in the US) in the introduction and then use LFS throughout*

**Done.**

*- One of the most challenging things about the current manuscript is that different phrases often get used for the same concepts. I encourage the authors to give each smoke exposure metric a single name and to rigorously use that name consistently throughout. For example: Facebook posts; Google searches; PM2.5 measurements (specifying AQS or IMPROVE where necessary); MODIS AOD; HMS plumes; and WRF-Chem PM2.5. Further, suggest that all of these metrics go under single Smoke Exposure Metrics subheading in the Methods section, and that each gets its own paragraph with its name in bold at the beginning. This will more easily help readers to refer back to the methods while they are pondering the results.*

**We have attempted to be more consistent in our terms. For example, we change all references of HMS (smoke, plume, product, etc.) to "HMS smoke product", all AOD references to "MODIS AOD", all WRF-Chem references to "WRF-Chem PM$_{2.5}$", and all Facebook references to "Facebook posters". We have also given each metric its own subheading in the Methods section.**

*Also, be clear up front that analyses were done at the 24-hour time scale.*

**We added this to the abstract.**

*- Methods section currently gives no information about how Facebook posts were compared with other metrics (temporal correlations, spatial correlations, etc). Much of this information is erroneously included in the Results. Suggest two subheadings be added to Methods: (1) Comparison of Facebook Posts with Conventional Metrics, and (2) Comparison of Other Metrics with PM2.5 Measurements. Main conclusions are that (1) Facebook Posts are correlated with conventional metrics, and (2) they are as correlated with the gold standard as other metrics. As such, suggest they clearly frame the Methods so that the conclusions naturally follow.*

**We have edited the first sentence in the methods section on surface observations to: "We determined the temporal correlation of these datasets to several other datasets that are commonly used for estimating exposure to wildland-fire smoke on a daily timescale." We have also added sentences describing the comparison to each subsection under Methods.**

*- Overall, suggest the following subheadings for Methods: Smoke Exposure Metrics; Comparison of Facebook Posts with Conventional Metrics; Comparison of Other Metrics with PM2.5 Measurements; Assessing the Effects of Cloudy Days; Regression Case Study in Washington State; and Using Google Searches to Evaluate Keyword Utility.*
**We have added several subheadings to the Methods section (although not the exact ones suggested) and put the means of comparison into the Methods section.**

*- As such, suggest the following subheadings for Results and Discussion: Facebook Posts Compared with Conventional Metrics; Other Metrics compared with PM2.5 Measurements; Cloudy Day Modification; Regression Case Study; and Keyword Utility for Smoke Detection.*
**We added several of these subheading suggestions to the manuscript. Results now has the following sections:**
**3.1 Comparison of Percent of Facebook Posters to Conventional Metrics**
**3.2 Evaluation of All Metrics Compared to Surface Measurements**
**3.3 Cloudy Day Modification**
**3.4 Google Trends Comparison with Surface Measurements**
**3.5 Google Trends Search Term Comparison**
**3.6 Geographically Weighted Regression Test Case for Washington state**

*It seems like there should be at least one table allowing readers to compare correlations between the main metrics (Google searches omitted).*
**The results are different for each site, and this spatial variability is important. We chose to show the results in Figure 4 rather than a table with the statistics for each site.**

*- Were cloudy days controlled for in the regression analysis? Sounds like they should be, given the findings of that sub analysis.*
**The cloudy days were not accounted for in the regression. We agree that they should be in future work (likely by including the cloud information into the regression model). However, this was just a first test and more work on the regression analysis is needed. We have added the following sentence to the discussion section in 3.3: "We also did not account for cloudy days in our regression analysis. Including information on cloud cover could potentially improve our regression model, which will be investigated further in ongoing work on this analysis."**

*Finally (and I know this is long – I'm sorry!), a good paper should stand alone without its figures and tables just as the figures and tables should stand alone without the paper. The authors most often use statements such as "Agreement between MODIS AOD and Facebook posts are shown in Figure 3" where they should use "Agreement between MODIS AOD and Facebook posts was moderate (Figure 3)". Similarly, "In Figure 2 we also show example time*

*series of Facebook posts" should be "An example of the time series of Facebook posts and other metrics shows that…(Figure 2)".*

**We respectfully disagree that papers should stand alone without figures and tables. Additionally, we think this is a stylistic decision to always introduce a figure and its overall meaning in the same initial sentence. We have not found an instance in the paper where we mention a figure without explaining what the figure suggests in the text. For example, the sentence "In Figure 2.." is followed by a second sentence (and several paragraphs) specifically discussing what the figure shows (combing the two sentences would make another long, complex sentence). We did change the other sentence to "Agreement between MODIS AOD and Facebook posting trends are shown in Figure 3c, which also shows the best agreement in the northwestern US".**

*Minor concerns:*
*- The statement about cloud cover in the abstract is not put in enough context to make sense. For example, one would assume that Facebook compares poorly with AOD on cloudy days because AOD performs poorly on cloudy days -- not because Facebook users (who have noses) perform poorly on cloudy days. The authors do present intriguing evidence to the contrary in the results, but this statement in the abstract shakes their credibility.*
**The sentence has been removed.**

*- There's a lot of weird and insistent use of hyphens in the text. For example air-quality is not conventionally hyphenated. Please review carefully and correct for common usage.*
**This comment was addressed by Dr. Pierce in another comment on hyphenation of compound adjectives. We have chosen to follow the grammatical rule rather than common usage.**

*- Paragraph numbed 15-25 on Page 3 has some really long and complex sentences. Please break up for more clarity.*
**We have rewritten the sentence as three separate sentences:**

**"Studies of health impacts often rely on (I) fixed-site monitors (e.g. Pope et al., 2009), (II) satellite products (e.g. Henderson et al., 2011; Rappold et al., 2011), or (III) atmospheric model simulations (Alman et al., 2016; Fann et al., 2012; Johnston et al., 2012; Rappold et al., 2012). Each of these methods has limitations as an exposure metric. For example, fixed site monitors are sparse in much of the western US, and satellite products do not on their own provide surface-level concentrations. Atmospheric model simulations may be biased by their emission inventories (Davis et al., 2015; Zhang et al., 2014), spatial resolution (Misenis and Zhang, 2010; Punger and West, 2013; Thompson et al., 2014; Thompson and Selin, 2012), or input meteorological fields (Cuchiara et al., 2014; Srinivas et al., 2015; Žabkar et al., 2013)."**

*- Referencing still using numbers rather than names in some places (line 23 on Page 3 and others).*
**This was addressed in the first author comment on the article. We did find one more, however on page 8, which should be** "(MEGAN, Guenther et al., 2006)".

*- There's a lot of use of the word "determine" which is quite strong. Its definition implies exactitude. Suggest words like "assess" and "evaluate" are more appropriate. Recommond title be changed to "… : Using social media to assess population smoke exposure".*
**Title is changed and several instances of "determine" been replaced by "evaluate" or "assess".**

*- AOD and MODIS used before their definitions.* **This has been fixed.**

*- Page 4: what are "air quality exposure" and "risk exposure assessment"? Do they mean "air pollution exposure" and "exposure assessment"?*
**Apologies this should be (and has been changed in the text) "risk and exposure assessment."**

*- Social media = plural, treated as singular* **We found and made one correction.**
*- Data = plural, treated as singular* **We found and made 3 corrections.**

*- Methods for population weighting / gridded estimates on Page 5 not very clearly described and nor is the rationale. Suggest statement about why this needs to be done and then described as a weighted spatial interpolation (which I think it is).*
**We have re-arranged the paragraph on gridding and weighting the raw data and added a few sentences. "We translate the raw Facebook data to a standard latitude/longitude grid using an area-smoothing procedure with data weighted by the population of the municipality (See Supplementary Figure 2 for example). The spatial interpolation allows us to estimate the magnitude of the response between the specific locations (centroids) and to compare to other gridded datasets. Additionally, we chose to weight the results by population because some of these locations are in areas with small populations (and potentially few posters on Facebook), which could skew our results."**

*- US or U.S. – choose one.*
**Done, US**

*- Page 12 brings up the question of visibility, for which the US has good data. Do not suggest that the authors do further analyses, but do suggest that they give thought to what such analyses could help to elucidate.*
**There are different measurements of visibility in the US, relating to either clouds/fog or aerosol concentrations (and water uptake). ASOS/AWOS visibility measurements at airports are for surface visibility (air clarity) and are given in statute miles. These measurements are airport-specific and not necessarily regionally representative. The IMPROVE network is used for visibility in National Parks as related to the Regional Haze Rule. We included measurements from the network, although we used mass concentration and not visual range or extinction.**

*- Pages 12 and 13, both line 15: because, not since*
**Changed.**

*- Figure 1: Suggest showing HMS plumes and PM2.5 Measurements in separate plots, just to make it really clear that we are dealing with five exposure metrics.*
**Done.**

*- Figure 2 is…er…a lot. Reducing to population-weighted only Facebook posts would help. Do you really need cloudy days on here? Can you do it with something other than dogs, if so? Why these four locations? Should be described in Methods.*
**We removed the unweighted (raw) Facebook from Figure 2 and moved it to Supplement Figure 3. We also removed the diamonds indicating cloudy days from Figure 2.**

**We chose four different locations as examples for different discussion points.**
    **1.) We include Pinehurst, ID because it was near the fire and has a low population. We show the impact of population weighting the Facebook data on Pinehurst compared to the three other locations which have larger populations, where population weighting made little difference on the resulting time series.**
    **2.) We chose Fort Collins to contrast downwind location compared to near-fire locations such as Bellingham and Great Falls.**
    **3.) We also chose Fort Collins and Great Falls because they had similar cloud cover for the time period to also contrast downwind location compared to near fire locations.**
**We added a sentence to the Methods and to the following sentence to the paragraph on Page 9 introducing Figure 2 in order to explain our reasoning: "All of these locations were impacted by wildfire smoke during the study period, but the response in the Facebook dataset varied among the sites likely due to differences in surface concentrations, distance to fire, population, and cloud cover."**

*- Time series is two words*
**Changed.**

Reviewer 2:
*In this paper the authors explore the power of social media data to improve data coverage on smoke exposure. As the need for increased data density in atmospheric exposure generally progresses, it is highly likely that more studies will rely on the 'citizen sensors' approach. I find the study a refreshing addition to the often stagnant observation based literature. It adds to the already wealthy cross-disciplinary arm of ACP and I enjoyed reading it.*
*I would like to see this published in ACP after some comments are addressed below.*
**Thank you, Dr. Topping, for your review and positive comments.**

*Could you remove multiple contributions from a particular individual from Facebook?*
**The value here is the % of Facebook posters rather than the % of posts on a given day. So, it does not include multiple contributions from an individual on the same day. We have clarified this in the text:**
**"The search generates de-identified and aggregated counts of posters each day, divided by the number of people who used Facebook in that city. This method counts each person at most once per day, avoiding bias from a single person posting multiple times about air quality that day."**

*How do you remove a particular biased commentary from a subset of users? I was interested in the lack of threshold PM2.5 concentration for people to start posting. Could this be a factor? If a small number of users are relying on available monitoring data, then reporting this, they might be driving a wider response. This isn't necessarily a negative feature, of course, but has parallels in social media coverage of viral outbreaks.*

**Because we are using aggregated data, we are unable to remove comments from a subset of users. We attempted to remove some false positives in our search by eliminating posts that mentioned certain terms (see Supplement), but these could still be a factor. Our search also does not include re-shares of news articles or friends' posts, so it would prevent a bias from a lot of people simply re-sharing the same post (i.e., "viral posts"). However, we are unable to tell if people are relying on actual monitor data, relating personal observations, or just repeating what they read from another post.**

*What percentage of facebook users are you actually obtaining? For example, twitter restricts access to a small percentage unless a fee is paid. Could you add this information to the manuscript?*

**We have all Facebook posters included in the aggregate value. Facebook does not sell posts; the search was conducted internally at Facebook by a Facebook data scientist. We clarified the text to state that it is all posts: "The search generates de-identified and aggregated data from all posts".**

*I often wonder how much an individual response is due to reporting on a news item/political debate rather than commentary on conditions experienced at any point in time. As with some practices in sentiment analysis, it might be useful to analyse bigrams/trigrams for a given post. Is that data available?*

**N-gram analysis would likely be useful here, and it would be interesting to analyze people's sentiments about smoke as well (are they experiencing any health effects? Are they taking precautionary measures? Do they know the source?). However, per our data use agreement to protect the privacy of people using Facebook, we were only provided aggregate values for our search terms and do not have access to the actual text of the posts. Therefore, with the data provided, we cannot analyze bigrams/trigrams. While our results showed that the dataset was well-correlated in regions that experienced smoke, we believe that further refinement of the search criteria could likely improve the results.**

*I appreciate the difficulty in providing social media data, having been personally rejected from other journals on this commonly known technicality. Would it be possible to provide a little more detail on the process of Facebook data retrieval for those who might want to replicate a similar study at least?*

**The data retrieval is conducted internally at Facebook by a Facebook data scientist. Much like with health data, data are provided only as an aggregate to protect users' privacy and not identify individuals, and with a strict data use agreement that requires the data are used only for a specific approved analysis. To replicate the study, a data use agreement would have to be set up for the research institute (again, much like with health data). We have added this note to the Data Availability section.**

*Regarding the regression model, was there a particular reason to opt for linear combinations of predictor variables? I wonder if the accuracy of your technique might be increased by even a simple decision tree, or ensemble method, and additional variables. Using k-folds cross validation and variable selection this might generate a more widely applicable method.*

**We used a linear combination of the predictor variables following the work presented in Lassman et al. (2017). The reason for using a linear combination was to more intuitively understand the relative importance of each variable in the model, and how this varies spatially. However, the reviewer makes a good point this added layer of complexity may improve the applicability of the model; other studies (e.g. Reid et al. 2015) have used decision trees and gradient-boosting models to do this, and it was very effective. But these more complex statistical approaches can make the results harder to understand. Therefore, because this is a proof-of-concept paper, we decided to keep the regression tool comparatively simple and intuitive for now. We do plan on exploring additional variables and models, refining our methods, and using a more rigorous validation process in future work.**

**Reid, C. E., Jerrett, M., Petersen, M. L., Pfister, G. G., Morefield, P. E., Tager, I. B., Raffuse, S. M. and Balmes, J. R.: Spatiotemporal Prediction of Fine Particulate Matter During the 2008 Northern California Wildfires Using Machine Learning, Environ. Sci. Technol., 49(6), 3887–3896, doi:10.1021/es505846r, 2015.**

*A minor comment on the line: 'social media datasets could currently improve estimates without the costly investment of computer modeling.' I would add this really depends on the application. If you were to fit a multivariate regression model to actual post content, with access to many hundreds of thousands of posts, the time to train a model varies with amount of data used. I leave it to the authors to decide on whether to retain this.*

**We removed "computer modeling and" from the text.**